# Identification and Evaluation of Tools Utilised for Measuring Food Provision in Childcare Centres and Primary Schools: A Systematic Review

**DOI:** 10.3390/ijerph19074096

**Published:** 2022-03-30

**Authors:** Audrey Elford, Cherice Gwee, Maliney Veal, Rati Jani, Ros Sambell, Shabnam Kashef, Penelope Love

**Affiliations:** 1School of Exercise and Nutrition Sciences, Institute for Physical Activity and Nutrition, Deakin University, Geelong, VIC 3216, Australia; elforda@deakin.edu.au; 2Faculty of Health, University of Canberra, Bruce, ACT 2617, Australia; u3125198@uni.canberra.edu.au (C.G.); u3199404@uni.canberra.edu.au (M.V.); 3School of Health Sciences and Social Work, Griffith University, Gold Coast, QLD 4222, Australia; r.jani@griffith.edu.au; 4School of Medical and Health Sciences, Nutrition and Health Innovation Research Institute, Edith Cowan University, Perth, WA 6027, Australia; r.sambell@ecu.edu.au; 5College of Nursing and Health Sciences, Flinders University, Adelaide, SA 5042, Australia; shabnam.kashef@flinders.edu.au

**Keywords:** childcare, primary school, food provision measurement, weighed food, menu review

## Abstract

Background: Children aged 2–11 years spend significant hours per week in early childhood education and care (ECEC) and primary schools. Whilst considered important environments to influence children’s food intake, there is heterogeneity in the tools utilised to assess food provision in these settings. This systematic review aimed to identify and evaluate tools used to measure food provision in ECEC and primary schools. Methods: The Preferred Reporting Items for Systematic Reviews (PRISMA) was followed. Publications (2003–2020) that implemented, validated, or developed measurement tools to assess food provision within ECEC or primary schools were included. Two reviewers extracted and evaluated studies, cross checked by a third reviewer and verified by all authors. The Academy of Nutrition and Dietetics Quality Criteria Checklist (QCC) was used to critically appraise each study. Results: Eighty-two studies were included in the review. Seven measurement tools were identified, namely, Menu review; Observation; Weighed food protocol; Questionnaire/survey; Digital photography; Quick menu audit; and Web-based menu assessment. An evidence-based evaluation was conducted for each tool. Conclusions: The weighed food protocol was found to be the most popular and accurate measurement tool to assess individual-level intake. Future research is recommended to develop and validate a tool to assess service-level food provision.

## 1. Introduction

Early childhood provides a unique window of opportunity to influence nutrition and dietary habits, as this is when food preferences and habits are formed, often tracking into adolescence and adulthood [1,2,3] and influencing health outcomes throughout the life course, in particular the risk of developing obesity [4]. Dietary patterns in High-middle income countries (HMIC) indicate that children’s dietary intakes do not meet nutrition guidelines, with an overall under-consumption of the core food groups, particularly vegetables and wholegrains, and over-consumption of discretionary foods, defined as processed foods high in fat, sugar and sodium [5,6,7,8]. According to the Global Burden of Disease Study across 195 countries, suboptimal dietary habits, (low intakes of wholegrains, fruit and vegetables and high intakes of sodium fat and sugar) account for more deaths than any other risk factor [9]. In addition, according to the World Health Organisation Global Strategy on Diet, Physical Activity and Health, the abovementioned dietary patterns, alongside sedentary behaviour, are the two main modifiable risks in the development of childhood overweight and obesity [10]. 

Instilling healthy dietary habits in childhood is therefore an important focus for public health interventions. While the home and family environments are regarded as primary settings to influence the dietary habits of children, early childhood education and care (ECEC) and primary school have become important environments for early intervention due to the significant time children spend in these settings. Approximately 50% of 3–5 year old children in HMIC countries are enrolled in ECEC [11,12] and almost 90% of children aged 6–11 years worldwide are enrolled in primary school [13,14]. Children enrolled in ECEC programmes in HMIC countries can spend up to five 8-h days each week in these settings [13,15,16], with average attendance rates being around 30 h per week in some countries [17,18]. Similarly, children enrolled in primary school, inclusive of Kindergarten up to Grade 6 [19], can spend up to five 7-h days per week in this setting [20,21]. Children attending ECEC and primary school therefore have a high level of exposure to external food environments for prolonged periods of time [15,22,23,24].

Despite the recognition that ECEC and primary school settings offer extensive reach for the promotion of healthy dietary habits, recent studies have identified suboptimal food provision and dietary consumption within these environments [22,23]. Furthermore, food provision within these settings differs within and between countries, with some ECEC/schools providing meals prepared on-site (e.g., sit-down meal services) and others relying on children to bring their own food from home (e.g., packed lunchboxes). In some instances, schools may also provide the option of purchasing food from a school canteen [25,26,27]. From a policy and intervention development perspective, the differentiation between measuring ‘service-level’ food provision (i.e., children being provided with adequate quantity and quality to eat) and measuring ‘individual-level’ food consumption (i.e., children eating adequate serves of the food provided to them) is important, as each of these scenarios poses unique challenges to be addressed. For example, ‘individual-level’ measurements that identify children as not consuming enough may be related more to feeding practices and the eating environment, rather than sub-optimal food provision in the ECEC/school [28,29]. Conversely if ‘service-level’ measurements do identify inadequate food provision, staff training on menu planning in line with guidelines may need addressing [30].

While a gold standard for the evaluation of individual intakes, namely weighed food measures, has been established [31,32], a standardised, accurate method for assessing service-level food provision has not. The availability of an accurate measure of service-level food provision within ECEC/school settings would provide an opportunity to establish a best practice method to facilitate consistent measurement within these settings [33,34,35,36]. Additionally, it would add credibility and rigour to comparisons between study findings regarding menu compliance, food provision and food wastage [32,37]. Standardised measurement and comparable findings could also inform national policy, guidelines, and recommendations, to optimise nutrition environments for children within ECEC/school settings.

The aims of this systematic review are therefore to identify measurement tools utilised in previous research for the assessment of service-level food provision within ECEC and primary school settings, to evaluate the strengths and weaknesses of these tools in the context of service-level food provision, and to provide recommendations for the standardisation measurement of service-level food provision within these settings. 

## 2. Materials and Methods

This review follows the Preferred Reporting Items for Systematic Reviews (PRISMA) [38] protocol, which includes a 27-item checklist and a four phase flow chart ensuring rigour and quality (Figure 1). The protocol for this review is registered with the International Prospective Register of Systematic Reviews (PROSPERO) (registration: CRD42018109719).

### 2.1. Search Strategy

Five health databases (MEDLINE, Scopus, CINAHL, Cochrane Library and Web of Science) were searched for full-text English-language publications published between January 2003 and 18 November 2020. Search terms were developed in consultation with an experienced university librarian, to inform the development of a PICO (population, intervention, comparison, outcome) derived framework with related key search terms. Key terms included (“early childhood education and care” OR “childcare” OR “child care” OR “long day care cent*” OR “day care” OR “kindergarten” OR “K-6” OR “pre-school” OR “primary school” OR “elementary school”) AND (“measur*” OR “survey” OR “assess*” OR “evaluat*” OR “tool*”) AND (“food” OR “menu”) AND (“food provision” OR “food service” OR “food waste” OR “plate waste” OR “wastage” OR “menu compliance” OR “nutrition guidelines” OR “nutrition policy”). 

### 2.2. Selection Criteria

The search strategy for this review aimed to identify publications that implemented, validated and/or developed measurement tools that were used to assess menu compliance, food provision and food wastage at a service level within ECEC [defined as long day care (LDC), kindergarten and preschool services where food is provided onsite) and primary school (where food is provided through a food service or school canteens) settings. No limitations were applied to study design, and articles were eligible if they were published in English-language. Studies were excluded if they were not based in ECEC or primary school settings, did not measure menu compliance, food provision and/or food wastage at a service level, or simply measured individual dietary intakes or eating behaviours of children. 

### 2.3. Study Selection

The reviewers used Covidence, a Cochrane-developed software, to search and select relevant studies for both the title and abstract screening and full-text screening for inclusion. Following the initial search, two reviewers (Y.G., M.V.) independently screened the eligibility of titles and abstracts against the established inclusion/exclusion criteria, with further eligibility screening using full text. After each independent assessment, the two reviewers discussed individual perspectives for each study, and if no consensus could be made, both a third and fourth reviewer independently screened the relevant study to resolve the decision (R.J., S.K.). The first author (A.E.) reviewed all included studies and cross-checked interpretation of the findings. All procedures associated with study selection were completed in Covidence.

### 2.4. Data Extraction

Two reviewers (Y.G., M.V.) extracted information from the included studies into a study-developed data extraction table, cross checked by the first author (A.E.) and verified by all authors (Tables 1 and 2). Data extracted from each study included: (1): author, (2): year of study, (3): study design, (4): country of origin, (5): setting, (6): study aims, (7) measurement method, (8) limitations and strengths, and (9) quality appraisal. Where studies measured both food provision and intake, only the method utilised for food provision was reported. The Academy of Nutrition and Dietetics Quality Criteria Checklist (QCC) was used to critically appraise each study [39], as our review included multiple types of study designs and this tool comprehensively critiques original research across all types of study designs. The validity questions within the QCC focus on study research questions, participant selection protocol, comparability between study groups where applicable, handling of withdrawals, blinding of groups, where applicable, details on the intervention protocol, validity and reliability of measurement outcomes, data analytical approach, justification of study findings and biases. The outcome of this appraisal was either positive (+) (the report clearly addressed issues of inclusion/exclusion, bias, generalisability, analysis and data collection), neutral (Ø) (the report was neither exceptionally weak or exceptionally strong) or negative (−) (issues were not adequately addressed) [39]. Two authors (Y.G., M.V.) independently appraised each study, verified by a third and fourth author (R.J., A.E.) (Appendix A). Cohen’s kappa (κ) was calculated to assess inter-rater agreement, with values interpreted as: <0 no agreement, 0.00–0.20 slight, 0.21–0.40 fair, 0.41–0.60 moderate, 0.61–0.80 substantial and 0.81–1.00 perfect agreement [40]. Cohen’s kappa result was within substantial agreement (κ = 0.647; 95% CI: 0.558–0.736).

## 3. Results

### 3.1. Search Results

Database searching identified 1687 studies, with 964 studies remaining after removal of duplicates. After abstract screening, a total of 129 articles were identified for full-text screening with 47 articles excluded because they (a) did not measure food provision and/or compliance at a service level (*n* = 40), (b) were published in a language other than English (*n* = 4), (c) were not based within ECEC and/or primary school settings (*n* = 2) or (d) were other review articles (*n* = 1). In total, 82 articles were included in this systematic review (Figure 1).

### 3.2. Description and Quality Appraisal of Studies

Based on the Academy of Nutrition and Dietetics Quality Criteria Checklist (QCC), no studies in this review had a negative quality criteria rating, most (70%, *n* = 57) had a positive rating, with 30% (*n* = 25) having a neutral rating (Appendix A). Most included studies were cross-sectional (*n* = 69). Other study designs included randomized controlled trials (RCT) (*n* = 7), pre-post cohort studies (*n* = 3), quasi-RCT (*n* = 2), and cluster-RCT (*n* = 1). Most studies (86%) were conducted in countries with developed economies [41], namely United States (*n* = 32), Australia (*n* = 14), United Kingdom (*n* = 8), Canada (*n* = 4), Poland (*n* = 3), Ireland (*n* = 1), Norway (*n* = 1), New Zealand (*n* = 1), Belgium (*n* = 1), Denmark (*n* = 1), Sweden (*n* = 1), France (*n* = 1), Slovenia (*n* = 1), Portugal (*n* = 1). Twelve studies (14%) were conducted in countries with developing economies [41], namely Brazil (*n* = 4), Guatemala (*n* = 3), Mexico (*n* = 2), Ghana (*n* = 1), Iran (*n* = 1), China (*n* = 1). 

Forty-seven studies were conducted within the ECEC setting and 35 studies in the primary school setting. Across both settings, seven methods for measuring food provision were identified, namely: (1) Menu review; (2) Visual observation; (3) Weighed food protocol; (4) Questionnaire/survey; (5) Digital photography (6) Quick menu audit and (7) Web-based menu assessment. Figure 2 outlines the number of studies utilising each measurement method in each setting.

There were several differences in how these methods were utilised. Menu reviews were undertaken by either entering menu data into nutrition software to compare to guidelines, categorising menu items into food groups to compare to guidelines, listing foods available at each meal to compare to guidelines, or using a scoring tool to assess compliance with guidelines. Similarly visual observational data were either categorised into food groups or used nutrition software to assess against guidelines; weighed food protocol data were analysed via nutrition software; and digital photography categorised data into proportions of food groups to compare to guidelines. Questionnaires/surveys varied, generally asking questions about availability of foods on menus rather than actual served quantities. Quick menu audits classified food and drinks as consumed every day, sometimes or occasionally. Web-based menu assessment tools enabled data entry by staff at the setting (as opposed to researchers), producing a comparison of menu and recipe data to guidelines. Table 1 and Table 2 provide detailed information regarding categorisation of identified measurement methods for the ECEC setting (*n* = 47) and primary school setting (*n* = 35).

**Table 1 ijerph-19-04096-t001:** Summary of studies measuring food provision in ECEC settings (*n* = 47).

Authors	Year	Study Design	Country	Setting Characteristics	Study Aims	Measurement Methods	Notes/Critique	Quality Rating
Alves et al. [42]	2015	Cross-sectional	Brazil	5 LDC centres Age: 7–24 months	To investigate the compliance of actually served lunch menus with the nutritionist prescribed menus in public childcare centres	menu review ^1(b)^ observation ^2^	Only lunch observed, however over a long period (6 consecutive weeks). Actual food served compared to those on menus, with only 20% of food served. matching menus completely in the 1–2-year age group room, with none of the 0–12 month rooms providing what is on the menu. Trained nutrition students conducted observation, protocol of visual observation was not provided.	Positive (+)
Andreyeva et al. [43]	2018a	Cross-sectional	United States	Participants: 838 preschool children Age: 3–5 years	To assess the dietary quality of lunches in childcare centres	weighed food method ^3^, including plate waste observation ^2(a)^	Training provided to observers with nutrition knowledge which increases the quality of recorded data The observers overlapped on at least one child to assess inter-rater reliability Single day data collection may not reflect usual food consumption patterns	Neutral (Ø)
Andreyeva et al. [44]	2018b	Cross-sectional	United States	Participants: 343 LDC centres Age: not stated	To evaluate the food environment for preschool-age children in LDC centres and describes adherence to the nutrition regulations	self-reported questionnaire ^4^	Self report: risk of self-reporting bias or self-reporting error Actual menus or meals served not captured in this methodology Questionnaire adapted from a validated questionnaire	Neutral (Ø)
Ball et al. [45]	2007	Cross-sectional	United States	Participants: 1 childcare centre over 2 days Age: not stated	To describe the development and testing of an observational method used to assess the food served and consumed by children in a child-care setting	observation ^2^	Reliability/tool development study 5 trained observers documented foods served and consumed and statistical analysis determined level of agreement Mean intraclass coefficient of 0.99 shows high inter-rater reliability	Neutral (Ø)
Bell et al. [46]	2015a	Quasi- experimental RCT	Australia	Participants: 96 LDC provided menus at baseline and 102 at follow up	To determine the impact of an implementation intervention designed to introduce policies and practices supportive of healthy eating in centre-based child-care services	menu review ^1(b)^	Use of qualified dietitians to review the menus. Large cohort Actual food served or consumed not measured	Positive (+)
Bell at al. [47]	2015b	Pre-post cohort study	Australia	Participants: 236 children (baseline) and 232 children (follow-up) in 20 LDC centres Age: 2–4 years	To determine whether nutrition award scheme improves children’s food and nutrient intakes	weighed food method ^3^ included plate waste	Actual intake accurately measured by trained professionals Large sample size Detailed data on one day, however variety of food offered over a longer period not captured	Positive (+)
Benjamin-Neelon et al. [48]	2010	Cross-sectional	United States	Participants: 84 LDC centres Age: <6 years)	To compare menus with actual foods and beverages served to children in child-care centres	observation ^2^	52% of food served matched menu entirely 86.6% of foods items matched foods on menus Only one day observe—may not reflect usual match with menu	Positive (+)
Benjamin-Neelon et al. [49]	2013	Cross-sectional	Mexico	Participants: 54 daily menus from 142 LDC centres Age: 4–72 months	To assess the nutritional quality of foods and beverages listed on menus serving children in government-sponsored childcare centres	menu review ^1(a)(b)^	Foods listed in the menus may not reflect the actual foods served in the childcare centres 54 days’ menus can capture variety, however no analysis for food variety in the methodology	Positive (+)
Benjamin-Neelon et al. [50]	2015	Cross-sectional	United Kingdom	Participant: 851 LDC centres Age: not stated	To describe foods and beverages served in childcare centres, assess provider behaviours related to feeding, and compare these practices to national guidelines	self-reported questionnaire ^4^	The questionnaire was modified for the United Kingdom from 3 instruments of which 2 were tested for validity and 1 for reliability Unable to assess all foods and beverages served (only selected) and therefore unable to assess entire dietary quality	Positive (+)
Breck et al. [51]	2016	Cross-sectional	United States	Participants: 95 LDC centres Age: not stated	To evaluate the extent to which child-care centre menus correspond with food and beverage items served to children	observation ^2^	Trained observers observed meals on 2 days for 93 centres and 1 day at 2 centres. 87% of foods served matched foods on menus	Neutral (Ø)
Chiriquí et al. [52]	2020	Pre-post study	United States	Participants: 58 LDC centres Age: not stated	To identify changes in food and beverage practices LDC due to implementation of updated standards	self-reported questionnaire ^4^	Questionnaire addressed certain items on menus, for example “serving a fruit or vegetable as a component of a meal or snack once a day”, but did not address specifics on amounts of foods offered to children Self report: risk of self-reporting bias or self-reporting error	Positive (+)
Copeland et al. [53]	2013	Cross-sectional	United States	Participant: 258 LDC centres Age: not specified (up to 5 years)	To compare the nutritional quality of meals to snacks	menu review ^1(b)^	Actual food served, or intake not measured	Neutral (Ø)
Dave et al. [54]	2018	Cross-sectional	United States	Participants: 9 LDC centres Age: 3–5 years	To assess the agreement of posted menus with foods served to 3- to 5-year-old children	observation ^2(a)^	Trained observers utilised a tested valid and reliable direct observation protocol When taking acceptable substitutions into consideration, actual food served matched menus 94–100% Small sample size in only one city	Neutral (Ø)
Dixon et al. [55]	2016	Cross-sectional	United States	Participants: 630 children over 2 consecutive days Age: 3–4 years	To compare foods and beverages provided to and consumed by children at childcare centres	observation ^2(a)^ of foods served and foods consumed	One trained data collector observed all food served to children, another trained data collector observed all food consumed by 3 children Foods provided met 50% of daily intake however most foods consumed did not meet this guideline, outlining the importance of considering not only provision but also intake	Neutral (Ø)
Doak et al. [56]	2012	Cross-sectional	Guatemala	Participants: 4 LDC centres over 5 consecutive days Age: 3–6 years	To analyse the variety and diversity of dietary items and their different origins offered in childcare menus	weighed food method ^3^ (no plate waste measured in this study)	Higher accuracy as actual measurement of food provision was conducted rather than based on menu assessment Quality and variety of menus also assessed. Actual intake not assessed	Neutral (Ø)
Erinosho et al. [57]	2011	Cross-sectional	United States	Participants: 40 LDC centres (240 children observed) Age: 3–4 years	To evaluate nutrition practices of group childcare centres and to assess whether dietary intakes of children at these centres meet nutrition recommendations	observation ^2(b)^	Trained researchers observed 3 children at a time and recorded all food consumed from 8 a.m.–2 p.m. on an adjusted US Department of Agriculture food record form It is not clear whether the afternoon snack consumption was observed which may explain why less than 50% consumed 50% of daily intake	Positive (+)
Erinosho et al. [58]	2013	Cross-sectional	United States	Participants: 120 children (20 LDC centres) Age: 3–5 years	To assess the quality of foods and beverages offered to preschool children (3 to 5 years old) in childcare centres	observation ^2(a)(b)^ using a validated observational system	Two days of dietary observations were conducted Observe actual food intake instead of food provided Variety of food consumed not analysed	Positive (+)
Finch et al. [59]	2019	RCT	Australia	Participants: 44 LDC centres Age: not stated	To assess the effectiveness of an intervention including training, provision of written menu feedback, and printed resources on increasing childcare compliance with nutrition guidelines	menu review ^1(b)^	Additional information was obtained from cooks if the information was not adequately reported in the menu Menu assessment was done by qualified dietitian Diet quality was assessed by compliance with food group provision and no discretionary food on menus	Positive (+)
Fleischhacker et al. [60]	2006	Cross-sectional	United States	Participants: 6 LDC centres—menus analysed over 6 months (77 days) Age: 3–5 years	To assess types of food served at a childcare centre compared with centre’s monthly menus	observation ^2^	Food served over a long period (6 months) compared with menus	Positive (+)
Foster et al. [61]	2015	Cross-sectional	United States	Participants: 29 LDC centres Age: not stated	To assess nutrition and physical activity policies in rural childcare centres	self-reported questionnaire ^4^	Reliance on self-reported data Subjected to issues such as self-reporting bias, reporting error, and over- and under-reporting Use of validated survey form	Positive (+)
Frampton et al. [62]	2014	Cross-sectional	United States	Participants: 83 LDC centres Age: 3–4 years	To examine macro-/micronutrient content of childcare centre menus, compared to one third of dietary requirements	menu review ^1(a)^	Actual food served can be different from planned menu Centres picked at random—reducing risk of selection bias Actual recipes were not obtained, rather “standardised” recipes based on menu descriptions, making this less accurate	Neutral (Ø)
Gerritsen et al. [63]	2017	Cross-sectional	New Zealand	Participants: 57 LDC centres Age: not stated	To describe food provision and evaluate menus in childcare services	menu review ^1(b)^	The menu scoring system was adapted from other studies; no information about validation The scoring system assessed quantity, variety, and quality	Neutral (Ø)
Grady et al. [64]	2019	Cross-sectional	Australia	Participants: 69 LDC centres Age: not stated	To examine menu planning practices, menu compliance with dietary guidelines	menu review ^1(b)^	Menu reviewed by trained dietician Diet quality was assessed by compliance with food group provision and no discretionary food on menus	Positive (+)
Grady et al. [65]	2020	RCT	Australia	Participants: 54 LDC centres Age: not stated	To assess the effectiveness of a Web-based menu planning tool in increasing the number of food groups on childcare service menus that comply with dietary guidelines.	web-based menu assessment tool ^6^	Centres received access to the web-based menu tool and training on how to use it. There were improvements in provision of fruit, vegetables, dairy and meat, and reduction in discretionary food, but no improvement in full compliance goes guidelines The tool improved food provision but did not translate into full menu compliance	Positive (+)
Gurzo et al. [66]	2020	Cross-sectional	United States	Participants: 680 LDC centres Age: not stated	To compare food/ beverage provisions between child care sites	self-reported questionnaire ^4^	Validated questionnaire Self-reported data subjected to reported bias (over-reporting of practices considered favourable) The survey did not assess foods and beverages usually served or consumed	Positive (+)
Hasnin et al. [67]	2020	Cross-sectional	United States	Participants: 3 LDC centres (108 children) Age: 3 to 5 years	To assess whether LDCs were meeting the updated guidelines for lunch and whether foods consumed met guidelines	observation ^2(a)(b)^	Use of validated visual observation method in childcare setting Only lunches were observed, data does not reflect usual consumption	Positive (+)
Henderson et al. [68]	2011	Cross-sectional	United States	Participants: 200 LDC centres Age: 3–5 years	To develop and validate a self-administered survey to assess the nutrition and physical activity environment of child-care centres	self-reported questionnaire ^4^	Validation study—items on survey compared to items on the menu Amounts of foods not assessed, only whether certain items were on the menu and how often. Foods on menu were divided into “healthy foods” and “unhealthy foods” and Pearson’s correlation utilised to assess correlations with survey items. Moderate correlation between unhealthy food score and survey items (r = 0.260; *p* < 0.05), and healthy food score and survey (r = 0.266; *p* < 0.05) Compared to menus rather than actual food served	Positive (+)
Himberg-Sundet et al. [69]	2019	Cross-sectional	Norway	Participants: 73 kindergartens Age: 3–5 years	To explore the associations between the economic, political, sociocultural and physical environments in kindergartens, along with the frequency and variety of vegetables served, and the number of vegetables eaten	self-reported questionnaire ^5^ weighed food method ^3^ for vegetables only (over 5 days)	Questionnaire piloted but not validated Agreement between vegetables served on questionnaire and served/consumed (weighed record) were not reported	Neutral (Ø)
Jennings et al. [70]	2011	Cross-sectional	Ireland	Participants: 54 preschools Age: <1 and 1–5 years	To determine the nutritional support pre-school managers needed, and enhance existing pre-school nutritional training and practices	self-reported questionnaire ^4^ via telephone	The questionnaire covered some items on menus but did not address quantities or quality of food served Interview surveys are subjected to interviewer bias	Positive (+)
Lessard et al. [71]	2013	Cross-sectional	United States	Participants: 179 childcare centres Age: 3–4 years old	To examine compliance with regulations related to nutrition in childcare settings	self-reported questionnaire ^4^	Components of food served assessed, e.g., “wholegrains on menu”, but menus or food served not analysed. Questionnaire not validated	Positive (+)
Longo-Silva et al. [72]	2013	Cross-sectional	Brazil	Participants: 366 LDC centre children Age: 12–36 months	To assess menu quality and plate waste in public day care centres	weighed food method ^3^ not entered into software—used a diet quality tool to assess	No information on validity Whilst food served and plate waste measured, exact amounts consumed were not analysed against guidelines. High percentages of plate waste in this study	Positive (+)
Maalouf et al. [73]	2013	Cross-sectional	United States	Participants: 24 LDC centres Age: not stated	To describe the nutritional quality of foods served and the mealtime environment in childcare centres	menu review ^1(a)^ and observation ^2^	Centre visits were unannounced and a registered dietitian who completed training conducted the on-site observation	Positive (+)
Myszkowska-Ryciak et al. [74]	2018a	Cross-sectional	Poland	Participant: 706 kindergartens Age: not stated	To evaluate the compliance with mandatory nutrition recommendations in preschools	self-reported questionnaire ^4^ menu review ^1(b)^	Validated questionnaire The reliability of results from interviews was increased as the data were verified by menu assessment	Positive (+)
Myszkowska-Ryciak et al. [75]	2018b	Cross-sectional	Poland	Participant: 706 preschools Age: 3–6 years	To assess the nutritional value of menus served in preschools	menu review ^1(d)^	Actual amounts of food on menu not accessed, only whether a food item was on the menu, for example—vegetables served with every meal	Positive (+)
Myszkowska-Ryciak et al. [76]	2019	Pre-post study	Poland	Participant: 231 preschools Age: not stated	To evaluate the effectiveness of the multicomponent educational program for improving the nutritional value of preschools menus	menu review ^1(a)^	The guidelines in this study were to meet 70% of a child’s daily intake, significantly more than the expected 50% in other studies Actual intake not measured Not clearly stated how the inventory reports were obtained, e.g., whether it was a self-report	Positive (+)
Nicklas et al. [77]	2013	Cross-sectional	United States	Participants: 796 preschool children Age: not stated	To examine the variability of food portions served and consumed by preschool children	digital photography ^5^	Digital photography method was accurate and reliable compared to weighed food Only lunch was measured Digital photography method may be considered as intrusive to the typical lunch time environment/ consumption of lunch meals	Positive (+)
O’Halloran et al. [78]	2018	Cross-sectional	Australia	Participants: 7 LDC centres Age: 3–4 years	To determine the average amount of sodium provided in lunches and snacks and the average amount of sodium consumed at lunch among preschool children in LDC centre	weighed food method ^3^ digital photography ^5^	Actual consumption was estimated by calculating average serves from 3 plates minus plate waste from digital photography Small sample size Single day data collection may not reflect usual food consumption pattern	Positive (+)
Parker et al. [79]	2011	Cross-sectional	United Kingdom	Participants: 34 nurseries Age: <5 years	To explore nutrition and food provision in preschools	menu review ^1(a)(b)^	Low response rate (2 of 34 nurseries provided full recipes and menus; remainder provided either only menus without recipes or part menus (e.g., lunch only)	Positive (+)
Retondario et al. [80]	2016	Cross-sectional	Brazil	Participant: 4 LDC centres over 5 days Age: 7–36 months	To determine the nutritional composition of meals provided in LDC centres and to compare observed values with the recommendations	weighed food method ^3^ with laboratory calculations of nutrients (not using nutrition software)	High costs for laboratory nutrient analyses (not utilising software). Data collected over 5 days therefore variations could be considered.	Positive (+)
Romaine et al. [81]	2007	Cross-sectional	Canada	Participant: 28 LDC centres Age: needs of an active 4-year-old utilised for reference values	To determine the nutritional adequacy and quality of menus in LDC centres	menu review ^1(b)^	Utilised a menu scoring for quantity and quality to compare to guidelines Validity of menu scoring tool not determined	Neutral (Ø)
Sambell et al. [36]	2019	Cross-sectional	Australia	Participant: 30 LDC centres Age: 2–3 years old	To outline the process of data collection for the measurement and auditing of food provision and food waste at a service level	weighed food method ^3^	Consistency in training research assistance increases reliability Repeating weighing procedures may increase validity (time-consuming and kitchen space restraints) Potential for social desirability bias among centre staff Two days data collection does not measure variations, increasing number of days may increase transferability of the outcome	Positive (+)
Schwartz et al. [82]	2015	Cross-sectional	United States	Participant: 38 preschools Age: 2.5–5.7 years	To assess the nutritional quality of lunches served at LDC and examine compliance of current practices compared to proposed meal pattern recommendations	observation ^2(a)^	Based on a valid and reliable observational system for assessing dietary intakes in children in childcare settings Inter-rater reliability was assessed prior to data collection One day of data collection—does not capture regular practices	Positive (+)
Turner-McGrievy et al. [83]	2014	Cross-sectional	United states	Participant: 1 LDC large LDC serving 200 children—menus over 15 days reviewed Age: 6 weeks and older	To examine changes of preschool during the implementation of the new program standards using a survey and nutrient analysis of menus	menu review ^1(a)^	Small study sample—1 facility Menu review was used to determine whether there were statistically significant changes to menus before and after implementation of a nutrition program Actual intake not measured	Positive (+)
Vossenaar et al. [84]	2015	Cross-sectional	Guatamala	Set menu for community centres in Guatamala (40 days/8 weeks) Age: 2–7 years	To determine the nutrient adequacy and food sources of nutrients provided by the diet served in LDC	menu review ^1(a)^ not compared to childcare guidelines but 24-h guidelines	Menu review over 40 days captures variations and variation in menus were captured through statistical analysis Nutrients in food offered compared to 100% of daily requirements, not the recommended 50% recommended Actual food served or intake not measured	Positive (+)
Vossenaar et al. [85]	2011	Cross-sectional	Guatamala	Participant: 4 LDC centres Age: 3–6 years	To assess the nutritional content and contribution to recommended nutrient intakes of the menu offerings in LDC	observation ^2^ weighed food ^3^	Food weighed and observed over 5 days—can detect some variation Plate waste was considered Small sample size	Neutral (Ø)
Ward et al. [86]	2017	Cross-sectional	Canada	61 LDC centres over 2 consecutive days Age: 3–5 years	To compare food served in LDCs with the nutritional recommendations and compared the nutritional composition of lunches served	weighed food method ^3^ and digital photography ^5^ for recording measurements and meal composition	Limited to measuring lunch (main) meals Limited to two consecutive days	Positive (+)
Yoong et al. [87]	2019	Cluster RCT	Australia	Participant: 25 LDC centres (395 children) Age: 2–5 years	To assess the efficacy of a food service implementation intervention designed to increase provision of foods	menu review ^1(b)(c)^	Quantities of food as well as quality of menus analysed against guidelines Self-reported data were validated with on-site observation	Positive (+)

^1(a)^ Food items on the menu are extracted and/or analysed with nutrition analysis software to compare against setting specific guidelines. ^1(b)^ Food items in the menu were analysed into food groups and compared against setting specific guidelines. ^1(c)^ A scoring tool was utilised to assess menu compliance against setting specific guidelines. ^1(d)^ Menu reviewed and compared to a list of foods available on the menu, for example, vegetables in every meal, but actual amounts of foods on menu not assessed. ^2^ Observation of foods served by nutrition trained researcher/s and compared to posted menus for comparison of foods served to foods on menu. ^2(a)^ Observation by nutrition trained researcher/s and analysed with nutrition analysis software to compare against setting specific guidelines. ^2(b)^ Observation of foods served by nutrition trained researcher/s and food analysed into food groups to compare against setting specific guidelines. ^3^ Weighed food method—food served, (and in some cases plate waste measured to closest gram to calculate actual intakes). Data entered onto nutrition analysis software and compared against setting specific guidelines. ^4^ A questionnaire includes questions related to food provided, nutritional practices and the nutrition environment. Menu compared to setting specific guidelines. ^5^ Foods consumed were photographed with a digital camera mounted on a tripod with standardised measures for distance between lens and centre of meal plate and camera angle. The photographs were compared to photographs of weighed reference portions of the food to estimate the percentage of food served and consumed and then compared to guidelines. ^6^ Web-based instruments designed for centres to enter their menus and receive results comparing menus to guidelines.

**Table 2 ijerph-19-04096-t002:** Summary of studies measuring food provision in primary school settings (*n* = 35).

Authors	Year	Study Design	Country	Setting Characteristics	Study Aims	Measurement Methods	Notes/Critique	Quality Rating
Agbozo et al. [88]	2018	Cross-sectional	Ghana	Participants: 7 public school and 6 private schools over 5 days Age: 8.7 ± 2.6 (public); 8.1 ± 2.5 (private)	To assess the dietary diversity and nutrient composition of on-site school lunch and estimate the extent to which it met the RNI for children aged 3–12 years.	weighed food ^3^	Menus weighed over 5 days—can detect variety 3 Serves weighed and average of 3 serves entered into software	Positive (+)
Aghdam et al. [89]	2018	Quasi-RCT	Iran	Participants: 8 primary schools Age: 9.13 ± 1.23 (control); 10.19 ± 1.45 (intervention)	To investigate the effects of health promotion intervention on the school food buffets	menu review ^1(d)^	Checklist of healthy and unhealthy foods available in food buffet No information on validity or reliability of the checklist utilised	Positive (+)
Beets et al. [90]	2015	Cross-sectional	United States	Participants: after school programs of 20 primary schools over 4 consecutive days Age: not stated	To assess the types of snacks served, whether the snacks meet existing nutrition policies and their cost	observation ^2(b)^	Inter-rater reliability was used to assess the agreement in food consumption estimation with 97% agreement and kappa (κ = 0.89) Use of trained research staff in estimating snack consumption Use of on-site observation rather than menu-based analysis which gives an accurate representation of provided foods	Positive (+)
Beets et al. [91]	2017	RCT	United States	Participants: 20 primary schools Age: 7.9 ± 1.9 (control); 7.9 ± 1.8 (intervention)	To evaluate the 2-year changes in the types of foods and beverages served during a community-based intervention designed to achieve the Healthy Eating Standards	observation ^2(b)^	Inter-rater agreement of 113 observations (55% of all snacks served) was 98.4% (κ = 0.98) Training provided to observers which increases the quality of recorded data	Positive (+)
Davies et al. [92]	2008	Cross-sectional	United Kingdom	Participants: 149 primary schools Age: 4–12 years	To evaluate food portion sizes in primary school using direct assessment	weighed food ^3^ using food groups for analysis	Weighing food items individually can be time-consuming, labour intensive, and costly to implement Study done in 5 consecutive days—can detect variations	Positive (+)
DeKeyzer et al. [93]	2012	Cross-sectional	Belgium	Participants: 2 primary schools Age: 6–12 years	To determine the nutritional adequacy and acceptability to children of vegetarian lunches served on ‘Thursday Veggie Day’	menu review ^1(c)^	Menu review of vegetarian meals served once a week utilised a scoring tool with 3 components (1 point for each component—the 3 components focused on fat and fibre in the meals)	Neutral (Ø)
Farris et al. [94]	2014	Cross-sectional	United States	Participants: 3 primary schools over 5 days Age: not stated	To examine the nutritional quality of packed lunches compared with school lunches after the implementation of new school lunch standards	observation ^2(a)^	Use of observers with nutrition knowledge and training was provided which increases the accuracy and reliability in dietary observation High agreement for item identification (90.7%) and portion estimation (86.8%) 5 days of meal observation improves the result accuracy as it account for day-to-day variations in the nutritional quality of provided meals	Positive (+)
Gatenby [95]	2007	Cross-sectional	United Kingdom	Participants: 2 primary schools over 5 consecutive days Age: 9–10 years	To assess the nutritional content of the meals, including children’s actual intake	weighed food ^3^	Small sample size Data collection over 5 days can detect variations in meals served	Neutral (Ø)
Gougeon et al. [96]	2011	Cohort study	Canada	Participants: 1 primary school—159 lunches over 10 years Age: not stated	To describe dietary assessment process of 1 school meal program and the nutritional adequacy of the meals	weighed food ^3^	Small meal samples were measured in each year (*n* = 0–27) Result should be interpreted with certain caution as very minimal meal samples were collected from each school (one breakfast and lunch sample) Meal sample was provided by on-site nutrition coordinator which may be subjected to selection bias	Positive (+)
Gregoric et al. [97]	2015	Cross-sectional	Slovenia	Participants: 194 schools—menus reviewed: 24 school lunches reviewed over 5 days Age: not stated	To evaluate the extent of implementation of dietary guidelines in schools and present various monitoring systems	menu review ^1(c)^ weighed food method ^3^	Menu quality scoring system was adapted from other study and modified according to the study purposes Weighed food method was utilised for a smaller subset (120 school lunches)	Positive (+)
Haroun et al. [98]	2011a	Cross-sectional	United Kingdom	Participants: 6696 children (136 primary schools) over 5 consecutive days Age: 3–12 years	To assess lunchtime provision of food and drink primary schools and to assess both choices and consumption of food and drink by children	weighed food ^3^	Training was provided to fieldworkers on sampling and data collection methods, which included recording and weighing food and drink items provided at lunchtime Large, nationally representative sample	Positive (+)
Haroun et al. [99]	2011b	Cross-sectional	United Kingdom	Participants: 6696 children (136 primary schools) Age: 3–12 years	To evaluate the introduction of new standards for school lunches on the nutritional profile of food and drink items provided by schools and chosen by children at lunchtime	weighed food ^3^	This was a second analysis from the same data collection described in the previous study [100]	Neutral (Ø)
Huang et al. [100]	2017	Cross-sectional	China	Participants: 2936 primary schools Age: not stated	To evaluate the intake of food and nutrients among primary school students, and provide recommendations for new school lunch standards	menu review ^1(a)^	Only 3 days’ lunches were included in the menu review The portion sizes were assumed to be static (standard weight of sample meal was used to calculate food consumption), but it could vary plate by plate	Neutral (Ø)
Ishdorj et al. [101]	2016	Cross-sectional	United states	Participants: 3 primary schools over 30 days Age: not stated	To assess the nutrient content of vegetables offered and examine the relation between the overall nutrient density and the costs of nutrients offered and wasted before and after the changes in school meal standards	weighed food ^3^ with aggregated plate waste	5–10 servings of vegetables were weighed, followed by aggregated plate waste Measured over 30 days	Positive (+)
Joyce et al. [102]	2020	RCT	United States	Participants: 40 primary school children Age: not stated	To compare acceptability and feasibility of best practice with typical school lunches	weighed food ^3^	Meals were weighed before, and waste weighed after Based on validated methodology [103]	Positive (+)
Kenney et al. [104]	2015	Cross-sectional	United States	Participants: 111 primary school children Age: not stated	To test the criterion validity and cost of three unobtrusive visual estimation methods compared with a plate-weighing method	weighed food ^3^ observation ^2(a)^ digital photography ^5^	Validation study—visual observation and digital photography valid compared to weighed food Result demonstrated high intra-class correlations among the three visual estimation methods to weighed measures (>0.92 for all aspects except water consumption which was 0.48 for the visual observation) Time and costs for implementation were also assessed—visual observation being the lowest cost	Neutral (Ø)
Lassen et al. [105]	2019	Cross-sectional	Denmark	Participants: 680 primary schools Age: 5–11 years	To examine compliance with food service guidelines for hot meals as well as self-evaluated focus on food waste reduction across settings	self-reported questionnaire ^4^	Questionnaire validated with observation of actual meals served. Self-reported may lead to recall and social desirability biases. Subjective information on food waste may not reflect the realistic wastage of food. Questionnaire only available in Danish version	Positive (+)
Liz Martins et al. [106]	2014	Cross-sectional	Portugal	Participants: 471 primary school children Age: 9–10 years	To validate the visual estimation method for aggregated plate waste of main dish	weighed food ^3^ visual estimation method observation ^2^	Validation study Use of trained researcher in data collection Visual estimation on a 6-point scale was not as accurate as the weighing method	Positive (+)
Masis et al. [107]	2017	Cross-sectional	United States	Participants: 2 primary schools Age: not stated	To design a replicable training protocol for visual estimation of fruit and vegetable (FV) intake of kindergarten through second-grade students through digital photography of lunch trays	digital photography ^5^	Measurement method modified from previously validated study—ipads used for photography Intra class coefficients improved through 3 training sessions (0.86 (0.61 to 0.98) by 3rd training session. Low cost and easy to implement	Positive (+)
Morin et al. [108]	2012	Cross-sectional	Canada	Participants: 56 primary schools	To describe the food offered for lunch in the cafeteria service lines in primary schools	observation ^2(a)^	Observation checklist validated. Research assistants were of nutrition background Training on interview techniques and observational procedures were provided Description of food available, no nutritional interpretation against guidelines	Positive (+)
Myers et al. [109]	2019	Cross-sectional	Australia	Participant: 136 primary schools Age: not stated	To assess the compliance of school canteen menus with the policy in primary schools	quick menu audit ^7^	Short menu audit methodology with high level of agreement with the gold standard of canteen observations Less time consuming than a more comprehensive audit	Positive (+)
Nathan et al. [110]	2013	Cross-sectional	Australia	Participant: 42 primary school principals Age: not stated	To assess the validity of a self-report by the principal to assess healthy eating and physical activity environments in primary schools	questionnaire ^4^ observation ^2^	Validity study—validated against observation Kappa statistics found reasonable agreement between survey and observation (range −0.6–0.81) 70% of items had moderate agreement	Positive (+)
Nathan et al. [111]	2016	RCT	Australia	Participant: 53 primary schools Age: 5–12 years	To examine whether a theoretically designed, multi-strategy intervention was effective in increasing the implementation of a healthy canteen policy	quick menu audit ^7^	Good sample size Method validated in a previous study [112]	Positive (+)
Ohri-Vachaspati et al. [113]	2012	Cross-sectional	United States	Participant: 620 primary schools Age: not stated	To investigate the association between program participation and availability of fresh fruits, salads, and vegetables at lunch as reported by school	self-reported questionnaire ^4^	The questionnaire was modified from another study, not validated Survey responses have potential reporting biases (e.g., desirability and response biases) No subsequent observations were used to validate survey responses	Neutral (Ø)
Patterson et al. [114]	2013	Cross-sectional	Sweden	Participant: 86 primary schools Age: not stated	To develop a feasible, valid, reliable web-based instrument to objectively evaluate school meal quality in primary schools	web based menu assessment ^6^	Validation study Food based criteria focuses only on four nutrients (fat, iron, vitamin D and fibre) and not food groups Sensitivity ranged from 0.85 to 1, specificity from 0.45–1 and accuracy 0.67–1, therefore found to be a feasible instrument for self-assessment of menus	Positive (+)
Pearce et al. [115]	2011	RCT	United Kingdom	Participant: 136 primary schools Age: 4–12 years	To compare the key differences between school lunches and packed lunches after the implementation of standards for school lunch	weighed food ^3^	Food choices were recorded and weighed prior to consumption Nutrition analysis was conducted by trained nutritionist This study examined both home-packed lunch and school meals	Neutral (Ø)
Pearce et al. [116]	2013	Cross-sectional	United Kingdom	Participant: 136 primary schools over 5 consecutive days Age: 4–12 years	To determine changes in portion size of food served in primary school following the introduction of nutrient-based standards	weighed food ^3^	Large sample size Measuring over 5 days can assess variations	Neutral (Ø)
Perez-Ferrer et al. [117]	2018	Cross-sectional	Mexico	Participant: 645 primary school children in 99 schools Age: 10.1 ± 1.3 years	To analyse the compliance with nutrition standards for foods sold in schools and children’s school snacks	self-reported questionnaire ^4^ observation ^2(a)^	The questionnaire was tested for face validity Direct observation of foods chosen by children in canteens observed (4 children at a time) Large sample size	Positive (+)
Reilly et al. [113]	2016	Cross-sectional	Australia	Participant: 38 primary schools Age: 5–12 years	To assess the validity and direct cost of four methods to assess policy compliance: self-report via a computer-assisted telephone interview, comprehensive and quick menu audits by dietitians, compared with observations	quick menu audit ^7^	Validation study of four measures: quick menu audit, comprehensive menu audit, and self-report surveys. Quick menu audit had the highest agreement (84%) compared with observation (kappa rating = 0.68) Quick menu audit limited to regions that provide a canteen facility similar to Australian, New Zealand or Dutch schools. Quick menu audit lowest cost	Positive (+)
Reilly et al. [118]	2018	Pre-post cohort study	Australia	Participant: 168 primary schools at baseline and 157 at follow up Age: 5–12 years	To assess the potential effectiveness of an intervention in increasing the implementation of a healthy canteen policy	quick menu audit ^7^	Quick menu audit methodology utilised in this study was validated for healthy canteen policy compliance in a previous study based on colour coded (red, amber and green) products available in canteens [114]	Neutral (Ø)
Taylor et al. [119]	2014	Cross-sectional	United States	Participant: 2 primary schools Age: not stated	To test the reliability and validity of digital imaging (DI) and digital imaging with observation (DI+O) in assessing children’s FV consumption during school lunch digital photography ^6^	digital photography ^5^	Validation study—validated against weighed food measurement Tested for inter-rater reliability Digital imaging was found to be a reliable and valid method 96% agreement, Pearson’s correlation (r = 0.88–0.98) Trays not observed continuously throughout lunch period—unable to track any additions/removal of items Requires skilled/trained researchers for accurate measures. Small sample size—only 2 schools	Positive (+)
Turner et al. [120]	2016	Cross-sectional	United States	Participant: 4360 primary schools Age: not stated	To evaluate changes and disparities in school lunch characteristics from 2006–2007 to 2013–2014	self-reported questionnaire ^4^	Large sample size Survey data subject to social desirability bias or lack of knowledge among respondents. Survey did not allow enough detail to consider issues such as number of servings per week, offering foods versus serving foods (i.e., what students selected), or how much food was consumed	Positive (+)
Vieux et al. [121]	2018	Cross-sectional	France	Participant: 20 lunches served over 20 consecutive days Age: not stated	To assess the nutritional impact of complying with school food standards	menu review ^1(c)^	Voluntary collection i.e., not representative of school food service in France and findings cannot be generalised Estimated nutrient content inaccuracies.	Positive (+)
Weber et al. [122]	2010	Cross-sectional	Brazil	Participant: 511 primary school children—food measured over 4 weeks Age: 7–10 years	To assess the nutritional quality of prepared foods available to primary-school children	Observation ^2(a)^ with nutrient analysis completed in a laboratory	Study only involved one primary school, however over a 4 week period High cost for sending meal samples to laboratory for nutrient analysis	Neutral (Ø)
Woods et al. [34]	2014	Cross-sectional	Australia	Participant: 263 primary schools Age: not stated	To assess the compliance of school canteens with their state or territory canteen guidelines	menu review ^1(b)^ food items colour coded against recommendations	Online menus assessed Menu assessment methodology adapted from other studies, no information of validation	Positive (+)

^1(a)^ Food items on the menu are extracted and/or analysed with nutrition analysis software to compare against setting specific guidelines. ^1(b)^ Food items in the menu were analysed into food groups and compared against setting specific guidelines. ^1(c)^ A scoring tool was utilised to assess menu compliance against setting specific guidelines. ^1(d)^ Menu reviewed and compared to a list of foods available on the menu, for example, vegetables in every meal, but actual amounts of foods on menu not assessed. ^2^ Observation of foods served by nutrition trained researcher/s and compared to posted menus for comparison of foods served to foods on menu. ^2(a)^ Observation by nutrition trained researcher/s and analysed with nutrition analysis software to compare against setting specific guidelines. ^2(b)^ Observation of foods served by nutrition trained researcher/s and food analysed into food groups to compare against setting specific guidelines. ^3^ Weighed food method—food served, (and in some cases plate waste measured to closest gram to calculate actual intakes). Data entered onto nutrition analysis software and compared against setting specific guidelines. ^4^ A questionnaire includes questions related to food provided, nutritional practices and the nutrition environment. Menu compared to setting specific guidelines. ^5^ Foods consumed were photographed with a digital camera mounted on a tripod with standardised measures for distance between lens and centre of meal plate and camera angle. The photographs were compared to photographs of weighed reference portions of the food to estimate the percentage of food served and consumed and then compared to guidelines. ^6^ Web-based instruments designed for centres to enter their menus and receive results comparing menus to guidelines. ^7^ School canteen quick menu audit: This tool assigns product information and serve sizes for each item based on common canteen menu items, eliminating the need to obtain additional information from canteen managers.

Table 3 summarises an evidence-based evaluation of the included studies, which is described in further detail in the discussion.

Measurement methods differed in validation and accuracy. Some visual observation methods were either validated (*n* = 4 out of 17) [54,67,82,108], adjusted from validated methods against weighed food records (*n* = 1 out of 17) [54], or tested for reliability between the observers (*n* = 5 out of 17) [45,82,90,91,94]. Some questionnaires were validated (*n* = 6 out of 13) [61,66,68,74,105,110] or adapted from validated questionnaires against visual observations or menu reviews (*n* = 2 out of 15) [50,54]. Digital photography was validated against weighed food measures in two studies [104,119] or adapted from a validated method in one study [107]. One study used a quick menu audit validated against visual observation in an Australian school canteen setting [112]. Web-based menu self-assessment was validated against menu items in one study [114]. 

## 4. Discussion

To the authors’ knowledge, this is the first systematic review to identify and evaluate measurement methods used to assess food provision and menu compliance in ECEC and primary school settings. Overall, 70% of the studies in this review had a positive rating, assessed according to the Academy of Nutrition and Dietetics’ Quality Criteria Checklist (QCC) [39], with the remainder being assigned a neutral rating, due to lower scoring on validity screening questions. It is important to note that some of the validity screening questions in the QCC, such as bias in subject selection and blinding of subjects, were not applicable for a number of included studies as it is unclear whether participating services are biased towards better food provision and blinding to food provision assessment at a service level is not possible. Given this, the use of an unvalidated measurement method contributed to a lower study quality rating (QCC) score. This indicates that the quality rating of food provision research in ECEC and primary school settings could be strengthened through the utilisation of validated measurement tools and by improving internal validity in research studies. 

This discussion will outline the evidence, including strengths and weaknesses, for each of the seven food provision methods/tools identified across ECEC and primary school settings, with recommendations to inform future research and practice. 

### 4.1. Menu Reviews

Neary a quarter of studies (23%; *n* = 19), with 32% in ECEC and 11% in primary school, were found to use menu reviews as a food provision measurement method [31,34,37,43,46,47,48,56,58,62,63,64,66,67,71,72,74,80,83]. Menu reviews were usually conducted to determine the quantity of food, and in some cases quality of food (*n* = 6 out of 20) and variety of food served (*n* = 2 out of 20) compared to set standards or guidelines. Food quantity was either assessed by analysing all items on the menu and comparing to the guidelines, which required detailed recipes with exact quantities of each item in the recipe; or by analysing the menu based on a list of foods that are available on the menu, for example vegetables in every meal, without the analysis of the exact amounts of food. These differences in determining quantity would influence accuracy of menu review, as those studies that analysed based on a list of food items available, would not accurately determine amounts of food provided. 

On average, menu reviews were conducted over 2 weeks or more, compared to observations which were conducted for 1 day or more. Menu reviews therefore have a unique advantage of capturing average food provision, as well as analysis of quantity, quality and variety of food provided over time. Only two studies in the ECEC setting focused on quantity as well as quality and variety of menus [81,106]. A major disadvantage of menu reviews is that planned menus may not always reflect actual food provision. Four studies compared planned menus to actual food served, with varying results [42,48,51,60]. A cross-sectional study (*n* = 6 LDC centres, children aged 3–5 years) conducted by Fleischhacker et al. [60] in the United States found planned menus were inconsistently followed by the childcare centres, with only 28% of food served matching the planned menu. Similarly, Alves et al. [42] (*n* = 5 LDC centres, children aged 7–24 months) found only 20% of food served matched the planned menus. Conversely, Benjamin-Neelon et al. [49] (*n* = 84, children aged under 6) found an 86.6% match between food items served and planned menus, and Breck et al. [51] (95 LDC centres, children’s ages not stated) found an 87% match. While the studies that found higher matches between planned menus and actual food served were larger studies, they were also conducted over a shorter time of 1–2 days [48,51]. The studies with a lower match between planned menus and food served were conducted over a longer time period, with Alves et al. comparing data over a 6 week period [42] and Fleishhacker over 6 months [60]. 

This suggests that menu reviews may not be an accurate indicator of actual food provision in ECEC and primary school settings over a longer time. Moreover, menu reviews may be compromised if insufficient information is available such as portion sizes, types of foods (e.g., low, or high fat milk), and methods of preparation, leading to researchers being tasked with making assumptions and potentially adding to inaccuracy of reporting [63,74]. Menu reviews also rely on skilled professionals for menu coding and nutrient analysis [46,59,109]. Additionally, evaluation testing for validity and reliability for menu reviews appears to be lacking. To the authors’ knowledge, there are currently no validated menu review tools for the ECEC or primary school setting. 

### 4.2. Visual Observation

Twenty-one per cent (*n* = 17) of included studies used visual observation to assess food provision in ECEC (25%) and primary school (14%) settings. This method requires trained observers to visually estimate the amount of food served to (and in some studies also consumed by) children, including visual estimation of portion sizes of foods before and after consumption [125]. A major limitation of visual observation is that food provision is determined through estimation rather than calculating the exact amount [104,106], therefore this method is highly reliant on well trained observers and a standardised protocol for data collection. The approach varied across studies with one study using a 5-point scale where an untouched plate scored 5, if at least one bite was consumed scored 4, if three-quarters of the food remained scored 3, if half the food remained scored 2, if a quarter of the food remained scored 1, and if no food remained scored 0 [106].

Visual observation may be less costly and time-consuming to implement as only training of observers is required, with one validation study (*n* = 111 primary school children) reporting this method as being lower in cost compared to weighed food method and digital photography [104]. A limitation of this method is the number of children that can be observed at one time, which was commonly reported as 4 children per observer at a time in most ECEC and primary school studies, suggesting this method is better suited to small scale settings [42,45,54,60,67,85]. In contrast, larger cohort studies (between 20 and 95 ECEC or primary school settings) either only observed 1 meal (ECEC/schools) or snack (ECEC) [44,48,51,57,82], with data collection not representative of usual food provision. 

Visual observation demonstrated an intraclass correlation of >0.92, indicating excellent reliability, when compared with weighed food measurement in a validation study in the primary school setting [104]; however, only snack consumption was measured. Conversely, another validation study conducted in the primary school setting found poor correlation (5.5–24.7%) for visual observation compared to the weighed food method; however, this study used a point scale for observation rather than estimating portion sizes [106]. A high variation was also noted in terms of number of days of observation, ranging from one day to a few weeks [42,48,51,60]. Studies conducted over fewer days may reduce the ability to collect representative data of usual food provision by not capturing day to day variations and therefore may not accurately reflect actual food provision in the ECEC/primary school setting. This lack of a recommended study length to demonstrate usual food provision therefore requires further investigation.

### 4.3. Self-Reported Questionnaire/Survey

Self-reported questionnaires were used in 17% of included studies, mainly in the ECEC setting (11 studies, 23%) [32,38,47,51,53,54,55,56,59,63,76], with only three studies in the primary school setting [90,99,101]. Questionnaires mostly assessed compliance of nutrition policies, menus and/or feeding practices in relation to prescribed guidelines. About half of the questionnaires used were validated or adapted from a validated questionnaire [44,50,61,87], validating items against visual observation or menu reviews, with no validation studies using weighed food measurement. Studies using questionnaires had cohorts ranging between 29 and 4360 ECEC centres/primary schools, with most having a study cohort of 200 ECEC centres/primary schools or above (*n* = 7 out of 13) [51,53,59,90,98,101,104]. This may imply suitability for application in large-scale ECEC/primary school settings.

Questionnaires, however, can be associated with social desirability bias, linked with under- or over-reporting of certain foods [126]. Moreover, a respondent’s lack of knowledge regarding various nutrition practices may result in reporting errors [44], as reported by Reilly et al. [112] who found poor agreement between self-reported data and data collected from on-site observations. Studies using self-reported questionnaires mostly collected data about the provision of certain items, for instance “vegetables served at each meal”, without specific data on the type or number of vegetables served. Such data do not provide accurate detail on foods provided in ECEC or primary school settings. As a possible solution to this, three studies used multiple methods, such as a questionnaire alongside weighed food records [54], a questionnaire alongside a menu review [59] and a questionnaire alongside observation [101]. Completing a questionnaire alongside another valid and accurate measurement method warrants further investigation as this may allow researchers to triangulate data and more accurately determine food provision. However, associated time and financial costs need to be an integral part of determining the realistic application of this approach.

### 4.4. Weighed Food Protocol

Weighed food protocols were used by 27% of included studies (*n* = 22), with more frequent use in the primary school (37%) than the ECEC setting (19%). Weighed food protocols are considered “gold standard” for measuring individual-level food intake and, in some studies, have been adjusted for use in ECEC and primary school settings [36,127,128]. Studies using this method tended to have smaller cohorts, with most ECEC studies assessing between 2 and 30 centres, and with only 3 of 13 primary school studies assessing cohorts of over 100 schools [92,98,116]. The limited use of this method for larger cohorts may be due to the labour intensiveness of weighing food served at the individual (‘plate’) level. As a potential solution to this, Sambell et al. [36] developed a weighed food protocol for service-level food provision, based on the ‘gold standard’, measuring raw ingredients in the preparation phase, and using the average portion of 3 ‘plates’ served to children. This method appears less labour intensive to implement and in addition, there is little disruption to the children during mealtime as measurements are conducted in the kitchen/preparation area [36]. This protocol has yet to be validated against individual plate measures and should be a key research activity given the potential as a scalable option for larger cohort studies. Thirty eight percent of studies utilising this method received a neutral QCC rating, as the items for validity were not clearly articulated. Researchers using the weighed food protocol may be making an assumption that, as the ‘gold standard’, clear articulation of validity is not needed; however, this reporting needs to occur to strengthen the quality of such research studies.

Another limitation for several studies using the weighed food protocol is the time over which data can realistically be captured. The number of days over which food provision was analysed using this method varied between 1 and 5 days, with no studies measuring food provision for more than 5 days. Whilst 5 days can capture some variations in food provision, capturing seasonal variations needs further consideration. Research is therefore required to determine an acceptable length for data collection to ensure findings accurately represent food provision. It may be that data collection needs to occur over a designated number of days over several time points to better reflect the seasonality variation in menus and more accurately reflect food provision per se.

More recently, weighed food protocols have included a measure of weighed food waste, which can provide a more accurate calculation of food consumption data in addition to food provision data. At a service level, the aggregated plate waste method is considered more suitable for food provision studies compared to the individual plate waste method [129], where total amount of food consumed is calculated by deducting the total amount of food wasted from the total amount of food served (as average portions) and dividing this by the number of children. A validation study by Chapman et al. [130] found good agreement between individual and aggregated plate waste methods; however, both plate waste methods potentially underestimated vegetable consumption as some menu items, such as sandwich fillings, were not measured separately [129]. There is potential to combine aggregated plate waste with Sambell et al.’s method [36] to get an accurate measure at service level of food served, consumed, and wasted, but this will require further research and validation. Although aggregated plate waste methods may provide more accurate data, researchers are unable to identify which food groups make the largest contribution to food waste, and additional space is required for food waste collection [131,132]. Capturing food waste in addition to food served also has the potential to respond to an increased interest in the cost saving benefits of reducing food waste in ECEC and primary school settings and subsequent climate impacts [133]. Future research should aim to include food waste measurement when researching food provision, focusing on the validation and standardisation of both individual and aggregated plate waste methods as scalable options. 

### 4.5. Digital Photography

Digital photography is based on visual estimation of food images, which are taken according to a standardised distance and angle from the food served [104]. Five studies used digital photography to measure food provision and menu compliance in ECEC (*n* = 3) and primary schools (*n* = 2) [62,71,89,92,103]. A validation study conducted by Kenney et al. (*n* = 111 primary school children) [104] compared digital photography with the weighed food protocol to assess the accuracy, time and costs involved in this method. Digital photography had good agreement with the weighed food protocol regarding accuracy of estimating food consumption, and implementation cost was less than the weighed food protocol [104]. However, this validation study only examined snack food consumption which may not be generalisable to other types of meals with more food components, and may therefore also be more costly when applied across all meals served [104]. Taylor et al. (*n* = 2 primary schools) [120] validated digital photography in the primary school setting against weighed food and food waste measures and found good validity (Pearson’s correlation r = 0.91–0.96) and strong inter-rater reliability (0.92 (95% CI 0.90 to 0.94) in the assessment of fruit and vegetable consumption. The study highlighted slight underestimation of starting portions and waste of leafy greens (salads) [119]. 

As both the above studies focused on certain menu items (snacks, fruits, or vegetables), the use of digital photography in assessing mixed meals, defined as meals with more than one component (for instance a meat and vegetable stew), is unclear and may not be generalisable to all meal types. Furthermore, digital photography requires specialised equipment, may have high respondent burden due to interruptions during mealtime for photographs, and incurs human and resource costs capturing and analysing photographic data. One included study provided a possible solution to this by using iPads as a lower cost than digital cameras [107], supported by three training sessions with intra class coefficients improving through training sessions (0.86–0.98 by the 3rd training session). This review found that digital photography was used in smaller cohort studies, with the maximum number of ECEC centres assessed being seven centres. The use of this method for larger cohorts therefore warrants further research.

### 4.6. Quick Menu Audit

This method of measuring food provision was only used in primary school settings in countries where children have the option of bringing food from home or purchasing some or all their food from a school canteen [112,134]. Four included studies used the quick menu audit method to assess the healthfulness of items available for purchase in the school canteen [109,111,112,118], capturing product information and serve sizes for each item on the canteen menu, thereby eliminating the need to obtain additional information from canteen managers [112]. Foods and drinks were colour coded based on a classification of foods recommended for daily consumption (green), foods that should be consumed on some days (amber) and foods that that should be consumed only occasionally as they are highly processed, high in fat, sugar and/or sodium (red) [112,118]. This tool was validated against the visual observation method by nutrition trained researchers with good agreement (kappa = 0.68, 84% agreement) [112]. Studies using this method assessed cohorts between 53 and 168 schools, indicating that this measurement method may be suitable for larger settings and research studies. The tool, however, is only applicable to settings where food is available to be purchased/selected by children, and therefore may not be an applicable tool for ECEC or primary school settings where food is provided on-site. 

### 4.7. Web-Based Menu Assessment

Whilst a web-based menu assessment could be classified as a menu review method, it is considered a stand-alone tool within this review due to its unique ability to be used by staff and health professionals within the ECEC/primary school setting for menu planning. One validation study in a primary school setting [114] found the web-based menu assessment to have good agreement (Cohen’s cappa > 0.60) and reasonable reliability (intraclass correlation ranged from 0.33 to 0.99—fair to almost perfect) compared to an on-site menu review conducted by a nutrition researcher [114]. It is important to note that this study assessed nutrient components of meals (saturated fat, iron, vitamin D and fibre) rather than overall healthfulness of menus. In recent years, there has been a shift to promote foods instead of nutrients, as evidenced in Dietary Guidelines around the world [135]. At times, collecting nutrient provision is more relevant; however, with robust data collection methods, both food group and nutrient data could be obtained to support better translation for different sectors. 

In addition, as discussed earlier, a menu review can be compromised by several issues, many of which would affect web-based menu assessments, such as menus not reflecting actual food provision. There is potential, however, for web-based menu assessments to assist ECEC and primary school staff in the planning of their menus to meet recommended guidelines. In Australia, two government-funded web-based menu assessment tools, namely FoodChecker [136] and FeedAustralia [137], are available to the ECEC setting. These websites, however, are based on jurisdictional guidelines and therefore measure menu compliance against different parameters within an environment that promotes national dietary guidance [138], which poses a barrier to wider (national) uptake of such a tool.

A randomised clinical trial on the use of FeedAustralia’s menu planning tool and its impacts on food provision found that while no centres using the tool reached full menu compliance, use of the tool was associated with improved provision amongst most food groups [65]. This study, however, relied on self-reported menu data and observational child dietary intake data, rather than actual food provision at the service level, to determine compliance. The potential of web-based menu assessment tools to support menu planning and self-assessed menu compliance and enhance food provision in ECEC and primary school settings therefore warrants further investigation. 

### 4.8. Implications for Research and Practice

There is a fundamental premise that children need to be provided with adequate serves of recommended food groups if they are expected to consume adequate serves, and in both ECEC and primary school settings, the assessment of this needs to be conducted through service-level food provision measurement.

This systematic review aimed to identify current methods/tools utilised for determining food provision at the service level in ECEC and primary school settings, and to provide a recommendation on a standardised approach based on these findings. This review found various degrees of validity and accuracy of measurement tools, and of note, there were varied benchmarks against which tools were validated. Utilising a standard protocol for the measurement of service-level food provision could potentially enhance research rigour, allow for the accurate comparison of research findings as well as monitor changes over time more accurately.

The weighed food protocol is considered the most accurate measurement of individual-level food intake, and therefore the ‘gold standard’ [127,128]. While used most frequently to assess individual-level food provision, within ECEC and primary school settings, a service-level protocol, where each ingredient is weighed and recorded prior to the meal being prepared, has been adapted from the ‘gold standard’ and applied in the ECEC setting [36]. Future research should aim to validate this method for use in ECEC and primary school settings and explore its potential scalability for larger cohorts.

It is important to consider the differences in primary school food provision environments across countries, such as the United States and certain schools the United Kingdom [139] where food is provided to children, compared with Australia, New Zealand and the Netherlands, where children often have access to a school canteen to select and purchase items if food is not brought from home [111,134]. In the latter, a weighed food protocol may prove to be laborious and impractical to measure food provision as the proportion of food either purchased from the canteen or brought from home is unknown. In this scenario, a quick menu audit tool appears to offer a low cost, low burden, and validated tool to categorise the healthfulness of foods available for purchase at the school canteen in primary school settings [65,114].

Finally, a web-based menu assessment tool shows promise for ECEC and primary school settings in supporting self-directed menu planning, and evidence suggests that it does improve the menus of ECEC services [65]. Web-based menu assessment is subject to the same limitations as menu reviews in that it does not necessarily measure actual food provision. Future research should further investigate the uptake of web-based menu assessment tools to determine ease of access and usefulness, overall validity, and scalability. 

### 4.9. Strengths and Limitations

A key strength of this systematic review is that it is the first study to examine types of measurement methods/tools used to assess service-level food provision in ECEC and primary school settings. Furthermore, this review offers recommendations to inform research and practice, and to guide the development and use of a standardised approach for the measurement of service-level food provision in ECEC and primary school settings. The process of data extraction and screening was overseen and cross checked by multiple authors, and the quality of each method/tool was also critiqued, thereby increasing the robustness of the review process. This study is not without limitations. Research articles may have been missed as no hand searching of articles was done in the review process and references of all included studies were not included in the search strategy. This review may be subject to publication bias as only peer-reviewed published English language studies were included. Finally, this review focused specifically on identifying food provision measurement methods/tools used within ECEC and primary school settings, consequently, recommendations may not be generalisable to other settings.

## 5. Conclusions

This is the first systematic review to identify and critique methods/tools used to assess service-level food provision within ECEC and primary school settings. Seven methods/tools were identified, with varying degrees of validity and accuracy, and varied benchmarks for which validity was measured against. This illustrates the importance of developing a standardised tool to measure and assess service-level food provision and menu compliance in ECEC and primary school settings. The review found the weighed food protocol to be the most commonly used and most accurate tool to measure individual-level food intakes. The weighed food protocol has potential for adaption to measure food provision at a service level; however, future research will be needed, including validation. Validating a standardised weighed food protocol to measure food provision at a service level will allow for accurate comparison of findings across ECEC and primary school settings, providing reliable monitoring data and opportunities to enhance food provision. 

## Figures and Tables

**Figure 1 ijerph-19-04096-f001:**
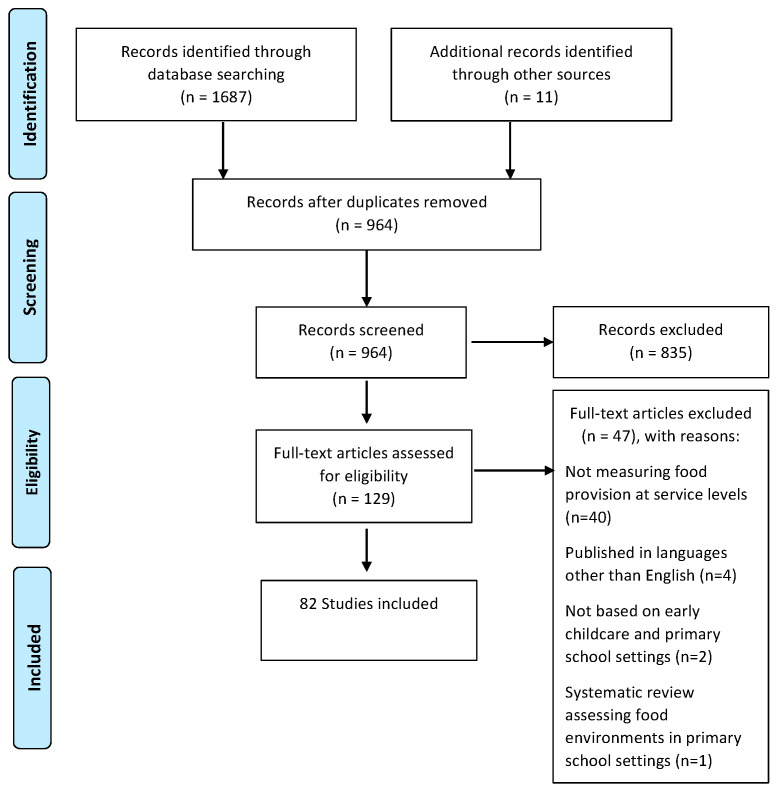
PRISMA flow chart.

**Figure 2 ijerph-19-04096-f002:**
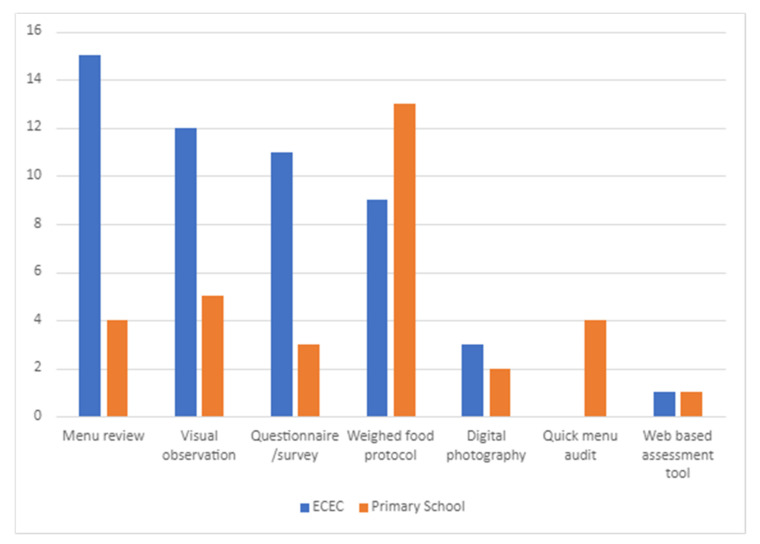
Identified measurement methods for food provision in ECEC and Primary School settings.

**Table 3 ijerph-19-04096-t003:** Evidence-based evaluation of measurement tools for assessing food provision and menu compliance in ECEC and primary school settings.

Method	Description	Evidence-Based Evaluation
Menu review	Food items on the menu are extracted and/or analysed with nutrition analysis software/divided into food groups/analysed per menu items/menu scoring tool created and compared against the dietary standards	2-week menu cycles or longer can assess variations in menus Mostly carried out by qualified dieticians/nutritionists [31,34,37,43,46,47,48,56,58,62,63,64,66,67,70,72,74,80,83] Differences found between menus and actual foods served [42,48,51,60] 73% of studies in this review (*n* = 15) had a positive QCC rating
Observation	Observation of foods served by trained researcher and compared to posted menus/analysed with nutrition analysis software to compare to guidelines/analysed into food groups to compare against guidelines.	Validated by Ball et al. [45] in ECEC setting (intraclass correlation coefficient: 0.99); applicable to all meal types served by the childcare centre [45] Validated by Kenney et al. [104] in the primary school setting (intraclass correlation > 0.92). This study evaluated visual observation and digital photography compared to weighed food records and found visual observation had lowest cost Visual estimation on a 6 point scale not as accurate as weighing method according to Liz Martins et al. [106] 64% of studies utilising this observation as a measuring tool (*n* = 11) had a positive QCC rating with the remaining 36% of studies having a neutral rating
Questionnaire/survey	Questionnaire/survey includes questions related to food provided, nutritional practices and the nutrition environment. Food provided are compared to guidelines.	Henderson et al. found moderate correlation between unhealthy/ healthy food score and survey items (r = 0.266; *p* < 0.05) [68] Does not measure food served but generally items on the menu or how often food is served Subjective method—open to desirability bias This method can assess large sample sizes Most studies (80%, *n* = 12) utilising this methodology had positive rating according to the QCC
Weighed food protocol	Food served, (and in some cases plate waste) measured to closest gram to calculate actual food served. Data entered onto nutrition analysis software and compared against setting specific guidelines.	Gold standard for measuring intakes [123,124] and has been adapted for accurately assessing food provision in ECEC [36] and primary school settings [106] Valid method [123], and whilst not validated in ECEC or primary school settings, studies of other measures utilises weighed food measures as a reference for validating measurement tools due to the accuracy of this method [77,104,106] Generally used in smaller sample sizes over a shorter period Eight out of 21 studies (38%) utilising this method received a neutral rating, due to criteria on validity being unclear in the study descriptions
Digital photography	Foods provided were photographed with a digital camera mounted on a tripod with standardised measures for distance between lens and centre of meal plate and camera angle. The photographs were compared to photographs of weighed reference portions of the food to estimate the percentage of food served and consumed and then compared to guidelines.	Validated tool in food consumption studies, but no validation studies for food provision at service level Validated by Kenney et al. [104] in a study examining food consumption in after school care facilities with good correlation between photography and weighed food methods (0.92–0.94) and good inter-rater reliability (0.84–0.95) Validated by Taylor et al. [119] against visual observation and found 96% agreement with an intraclass correlation of 0.92 Four out of the five studies that utilised this measurement tool had a positive QCC rating
Quick menu audit	This tool assigns product information and serve sizes for each item based on common canteen menu items, eliminating the need to obtain additional information from canteen managers. Foods and drinks are colour coded based on classification of every day (green), sometimes (orange) or occasional foods (red).	Specific tool for assessing primary school canteens in settings such as Australian schools where children can bring food/lunchboxes from home or purchase foods from their school canteen Not an appropriate measuring tool for ECEC or primary school settings where all food is served Validity study found agreement between quick menu audit tool and observations to be 84% with Kappa of 0.68 [112] Three out of the four studies that utilised this method had positive QCC ratings, with the fourth receiving a neutral rating due to some validity questions not clearly articulated in the studies
Web based menu self-assessment tool	Designed for centres to enter their menus and receive results comparing menus to guidelines.	Validation study conducted by Patterson et al. in a primary school setting [114] Sensitivity ranged from 0.85 to 1, specificity from 0.45–1.00 and accuracy 0.67–1.00, therefore found to be a feasible instrument for self-assessment of menus Clinical trial by Grady et al. [65] found that use of the tool by centres for self-measurement resulted in improvement in food group provision but not full compliance Both the studies utilising this method had a positive QCC rating

## Data Availability

Not applicable.

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
