# Peer review of "Identification and Evaluation of Tools Utilised for Measuring Food Provision in Childcare Centres and Primary Schools: A Systematic Review"

_ijerph, 2022, doi:10.3390/ijerph19074096_

Round 1

Reviewer 1 Report

Thank you for the opportunity to review the manuscript, “Identification and evaluation of tools utilised for measuring food provision in childcare centres and primary schools: A systematic review”. This systematic review provides a summary of tools used to assess food provision in childcare and primary school settings. The manuscript is well-written, and these findings are interesting with implications relevant for the journal’s readership; I have suggestions below to add clarity on the importance of primary aims, and to better integrate the Academy QCC tool into the Discussion.

Abstract

No comment, nicely written and concise.

Introduction

Lines 54-59, Please clarify whether each of these statistics are from specific countries only, developed countries, or are observed internationally.

Lines 64-68, This sentence is a little confusing. It reads that some countries are more likely to prepare food on site, and some more likely to have children bring from home. This might be the case but as it is, the sentence is ambiguous. Recommend revising to clarify the main message.

Lines 68-70, Recommend strengthening the argument that it is important to understand food provision at the service level, versus plate level, or children’s dietary intake. What does this service-level information add to the literature and what specific implications does it provide for future policy and intervention?

Line 75, This literature review proposes to identify “standardized measurement”, though a gold standard has already been established based on accuracy (i.e., food weighing). The authors should make it clearer here how summarizing what has been used in the previous literature adds to what we already know broadly on which assessment tools are most valid and reliable.

Methods

Line 90, Need to define “PICO”.

Lines 92-95 need spell checked, “long day care cent*”, kindergarten is missing quotations, childcare missing quotations, “measur*”, “evaluat*”.

Lines 126-127, Please explain why this specific quality criteria checklist was selected.

Lines 126-129, Highly recommend a summary of the QCC survey items, or what broad constructs they assess as being related to quality. The 10 items are listed in the Supplementary Material but should also be described briefly in the Methods, and those specific constructs should be tied into the study Results and Discussion.

Results

Line 149, Recommend including what percent of ECECs were center-based, home-based, Head Start, mixed sample, or other.

Line 241 & Table 3, Is this evaluation based on the Academy QCC? If so, that verbiage should be used consistently throughout.

Discussion

Generally, it is unclear throughout the Discussion how the Academy QCC was used to evaluate the studies. This reads more like a narrative review of the literature. As previously stated, would recommend consistent use of verbiage related to the QCC tool and its survey items or constructs.

Line 264, Please briefly clarify in this statement how “quantity” of foods was measured using menus; serving size is typically not listed on menus, but maybe in these studies they were required. Or, perhaps frequency of serving certain food items is more accurate?

Line 485, It seems to come out of no where that this statistic on children’s health is focused on Australia. Does this review aim to provide guidance for an Australian audience specifically? If so, this needs to be stated in the Introduction and it is unclear how this would be done with an international sample.

Lines 506-515, This is such a great summary of these findings and future implications! Very nice read.

Author Response

Many thanks for your comprehensive feedback on our manuscript.  All feedback has been addressed as outlined on the attached document.

Reviewer 2 Report

The review summarizes various measurement tools for evaluating children's nutrition in ECEC and primary schools. Consequently, it is not clear to me why the problem of childhood obesity is described in the introduction (lines 32-46). In my opinion, this is superfluous because it is not related to the purpose of the study.

The full name should be given when using the abbreviation for the first time. What does the abbreviation LDC mean?

Were any correlations observed between the measurement method and the quality rating in the study (Table 1 and 2)? It would be advisable to describe it in the publication, otherwise there is no real justification for the quality rating of the studies analyzed.

Thanks for the extensive research!

Author Response

Thank you for reviewing our manuscript.  All feedback has been addressed as outlined in the attached document.  
